# ADAPTIVE WAVELET TRANSFORMER NETWORK FOR 3D SHAPE REPRESENTATION LEARNING

**Hao Huang, Yi Fang**
NYU Multimedia and Visual Computing Lab, USA
NYUAD Center for Artificial Intelligence and Robotics (CAIR), Abu Dhabi, UAE
NYU Tandon School of Engineering, New York University, USA
New York University Abu Dhabi, UAE
{hh1811,yfang}@nyu.edu

## ABSTRACT

We present a novel method for 3D shape representation learning using multi-scale wavelet decomposition. Previous works often decompose 3D shapes into complementary components in spatial domain at a single scale. In this work, we study to decompose 3D shapes into sub-bands components in frequency domain at multiple scales, resulting in a hierarchical decomposition tree in a principled manner rooted in *multi-resolution wavelet analysis*. Specifically, we propose Adaptive Wavelet Transformer Network (AWT-Net) that firstly generates approximation or detail wavelet coefficients per point, classifying each point into high or low sub-bands components, using lifting scheme at multiple scales recursively and hierarchically. Then, AWT-Net exploits Transformer to enhance the original shape features by querying and fusing features from different but integrated sub-bands. The wavelet coefficients can be learned without direct supervision on coefficients, and AWT-Net is fully differentiable and can be learned in an end-to-end fashion. Extensive experiments demonstrate that AWT-Net achieves competitive performance on 3D shape classification and segmentation benchmarks.

## 1 INTRODUCTION

Analysis of 3D point cloud is a crucial topic in computer vision and graphics, and has wide applications in robotics (Chen et al., 2019), autonomous driving (Qi et al., 2018), and visual SLAM (Hitchcox & Forbes, 2020), *etc*. To better understand the 3D shapes, effective point cloud analysis approaches and methods are in great demand. With the thriving of deep learning, numerous task-driven neural networks are created to generate high-level 3D shape representations with learnable neurons, rather than designing low-level shape descriptors manually. Point-Net (Qi et al., 2017a) is firstly proposed to use a deep neural network to directly consumes take as input a

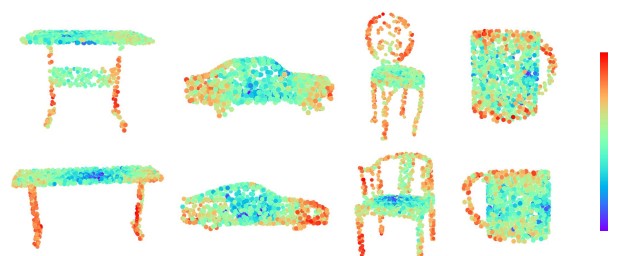

Figure 1: The detail component of table, car, chair, and cup from the test set of ModelNet40 (Wu et al., 2015). Colors represent the absolute values of the detail coefficients per point. The larger values (redder colors), the higher probability that the points locate on high-frequency parts, *e.g.*, edges or non-flat areas. Such geometric representations are learned implicitly in 3D shape classification task.

raw point cloud and max-pools per-point features to obtain the global shape representation. Point-Net++ (Qi et al., 2017b) introduces feature aggregation operations at multi-scales to group of the entire point clouds in both down- and up-sampling layers. The following works learn 3D shape representation from different perspectives: SO-Net (Li et al., 2018a) utilizes self-organizing map to model the spatial distribution of points, SpiderCNN (Xu et al., 2018) exploits step function and Taylor polynomial to captures local geodesic efficiently, KPConv (Thomas et al., 2019) designs de-

formable convolutions to localize kernel points for local geometry learning, DGCNN (Wang et al., 2019) treats point clouds as graphs and applies graph convolution networks, to name a few.

An interesting yet challenging problem is how to decompose 3D shapes into interpretable representations and exploit them for better shape analysis, *e.g.*, decomposing shapes into areas containing flat planes or areas containing edges. Another challenging problem is how to analyze 3D shapes at multiple scales/resolutions in an explainable way. Though some works (Qi et al., 2017b; Jiang et al., 2019) extract point cloud features hierarchically, these works build the hierarchy in spatial/semantic spaces based on pairwise point distance, which is less interpretable *w.r.t.* shape geometry.

As indicated in GDA-Net (Xu et al., 2021b), the geometric components with different frequencies in 3D shapes contain distinct geometric characteristics (*e.g.*, high-frequency components correspond to edges or sharp parts, while low-frequency components correspond to flat areas or smooth parts) and provide complementary geometric information for shape representation. To effectively learn 3D shape representation, it is desirable to separate these components and process them differently, rather than feed the whole shape into a single processor. GDA-Net (Xu et al., 2021b) disentangles 3D shapes into sharp and gentle variation components using spectral graph analysis. Generally, GDA-Net starts from building a dynamic graph from a given point cloud, then applies a 2-order polynomial approximation (Laplacian) as filter to the graph to disentangle the graph into high and low-frequency components, and finally fuses the two component features with the original shape features through two attention modules. GDA-Net has achieved excellent performance on 3D shape classification and segmentation. However, for a given graph, the spectral analysis based approach adopted in GDA-Net provides a static and deterministic graph decomposition, posing a limitation that disconnects the graph decomposition process from downstream tasks of shape representation learning. Therefore, to address the limitation, we propose a novel wavelet analysis based method to decompose 3D shapes into sub-bands components for 3D shape representation learning. The decomposition process is realized by neural networks in a learnable fashion, thus connected to the downstream tasks of shape representation learning. Theoretical properties of wavelets in signal processing, such as time-frequency localization and multi-resolution analysis, have been well studied, making it appropriate to fuse into deep neural networks with interpretability. Inspired by that, we propose to perform multi-resolution analysis within neural networks through lifting scheme (Sweldens, 1998) to achieve a data-driven wavelet transform. The multi-resolution analysis generates decomposed visual representation at each scale, which contributes to the interpretability of the network. The generated approximation or detail coefficients per point at each scale, corresponding to high- or low-frequency components, capture essential geometry for downstream tasks.

Specifically, we first adapt wavelet transform that analyzes 1D temporal or 2D image signals to 3D shapes at different semantic levels and decompose the original point cloud into the approximation/coarse and detail components, as explained in Section 4.1 and 4.2. Next, in Section 4.3, we utilize Transfomer (Vaswani et al., 2017) to pay different attention to features from approximation and detail variation components, and then fuse them with the original point features, respectively. As shown in Figure 1, the approximation and detail components have distinct but complementary effects on reflecting the geometry of 3D shapes. Equipped with wavelet transform and Transformer, we propose *Adaptive Wavelet Transformer Network* (AWT-Net) that captures and refines the integrated and complementary geometry of 3D shapes to supplement neighboring local information. AWT-Net is trained using the loss functions proposed in Section 4.4. Experimental results on challenging benchmarks demonstrate that our AWT-Net achieves competitive performance on 3D shape classification and segmentation tasks. Comprehensive analysis and visualizations verify that our model effectively extracts the complementary geometric features from two decomposed components at multiple scales. For instance, as shown in Figure 1, we observe that our model can learn meaningfully samplings of point clouds in different categories and can detect key regions consistently within different categories, *e.g.*, it separates chair legs and backs from chair seat, and separates a cup handle from a body.

## 2 RELATED WORK

**Geometric Point Cloud Models.** PointNet (Qi et al., 2017a) and DeepSet (Zaheer et al., 2017) are pioneering neural network models that directly process point cloud to learn a spatial encoding of each point and then aggregate all individual point features to an integrated representation. PointNet++ (Qi et al., 2017b) partitions point cloud into overlapping local regions by the distance metric

of spatial and/or semantic spaces and extracts local features capturing fine geometric structures from neighbors hierarchically. DGCNN (Wang et al., 2019) reconstructs k-NN graphs using nearest neighbors in feature space to capture local geometric structure while maintaining permutation invariance. KC-Net (Shen et al., 2018) proposes kernel correlation layers to extract the local geometric features. Point2Sequence (Liu et al., 2019a) designs a sequence model to capture the correlations among points. FPConv (Lin et al., 2020a) introduces a surface-style convolution operator to directly work on point cloud surface. Point-GNN (Shi & Rajkumar, 2020) learns local shape geometry by capturing interactions among points in local patches through node and edge feature aggregation. PAConv (Xu et al., 2021a) proposes a convolution kernel that adaptively assembles different weight matrices using coefficients learned from point positions. GDA-Net (Xu et al., 2021b) is built on graph spectral theory and uses a 2-order polynomial approximation (Laplacian) as a filter and applies it to a graph to disentangle the graph into high and low-frequency components, thus resulting in deterministic and static decomposition. Our model learns point cloud geometry from a novel perspective, *i.e.*, multi-resolution wavelet analysis, which has been extensively applied in signal processing but seldom explored for 3D shapes.

**Wavelet in Vision.** Theoretical properties of multi-resolution analysis using wavelets have been well studied, making such approaches more interpretable than CNNs. Several prior works incorporate wavelet representations into CNNs. (Oyallon et al., 2017) proposes a hybrid network that replaces the first layers of a ResNet with a wavelet scattering network. This modified ResNet resulted in comparable performance to the original ResNet but has fewer trainable parameters. The wavelet pooling layer proposed in (Williams & Li, 2018) subsamples image features based on second-level wavelet decomposition. Wavelet-SRNet (Huang et al., 2017) reconstructs super-resolution images from multi-scale wavelet coefficients estimated from low-resolution images under coefficient supervision. Similar scheme is adapted in (Liu et al., 2020) for image demoiréing and (Rong et al., 2020) for burst denoising. In (Ramamonjisoa et al., 2021), they decompose a single image using Haar Transform and then reconstruct depth images hierarchically through inverse discrete wavelet transform. All of the mentioned works use pre-defined Haar wavelet which is less flexible for complex 3D shapes. (Rustamov & Guibas, 2013) adopts lifting scheme but still restricts to linear transformations of Haar coefficients for graph signals. DAWN-Net (Rodriguez et al., 2020) uses lifting scheme to implicitly define wavelet transform functions for image recognition. Due to the irregularity of 3D shape point arrangement, applying lifting scheme to learn 3D shape representation is more challenging than to 2D images proposed in DAWN-Net, and as far as we know, we are among the first to tackle this problem.

**Transformer in Vision.** The seminal work to adapt Transformer (Vaswani et al., 2017) to vision tasks commences with Vision Transformer (ViT) (Dosovitskiy et al., 2020) and Detection Transformer (Carion et al., 2020). Since then, various applications of Transformer in vision are proposed, including but not limited to: semantic segmentation (Wang et al., 2021; Zhang et al., 2020), generative modeling (Chen et al., 2020; Hudson & Zitnick, 2021), object detection (Zhu et al., 2021; Dai et al., 2021), video analysis (Gabeur et al., 2020; Zeng et al., 2020), to name a few. PCT (Guo et al., 2021) applies Transformer in 3D point cloud classification and segmentation. Concurrent works include the other two types of Point Transfomer proposed by (Zhao et al., 2021) and (Engel et al., 2021). PoinTr (Yu et al., 2021) converts a point cloud to a sequence of point proxies and employs Transformer for point cloud completion. PPT-Net (Hui et al., 2021) adopts Transformer for large-scale place recognition. MViT (Fan et al., 2021) and Swin (Liu et al., 2021) connect multi-scale feature hierarchies with transformer models by applying different Transformers for features with distinct spatial resolutions. Since multi-scale discrete wavelet transform decomposes point clouds at each subsequent scales, it is intrinsically essential to combine with Transformer in a similar way as in MViT and Swin. Due to quadratic computational complexity of self-attention, Fast Autoregressive Transformer (Katharopoulos et al., 2020) is among the first to address the quadratic complexity by using the associativity property of matrix products. Some recent works (Kitaev et al., 2020; Dai et al., 2020; Tay et al., 2020; Choromanski et al., 2021) propose efficient Transformers for long sequences and these models can be also adapted for vision tasks.

## 3 BACKGROUND

We briefly review wavelet transform and lifting scheme as the building blocks of our model. We refer the reader to (Sweldens, 1996; 1998; Claypoole et al., 2003; Rodriguez et al., 2020) for more information.

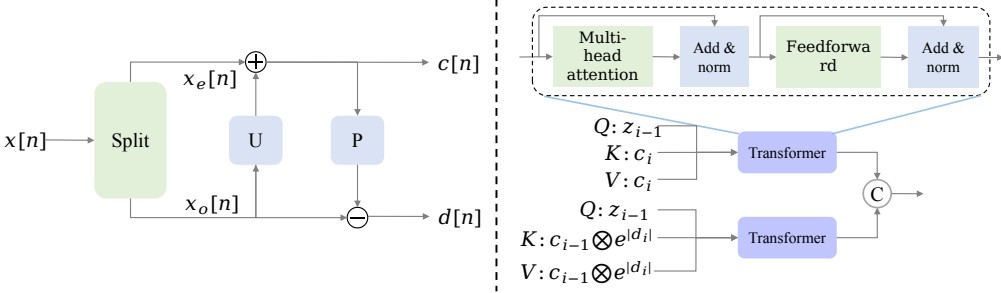

Figure 2: (Left) Lifting stages: split, update (U) and predict (P). (Right) The architecture of Transformer in AWT-Net.

**Wavelet Transform/Decomposition.** Let us consider a continuous scalar function of a single variable $f(x)$. Wavelet transform is to find a basis generated by a mother wavelet, denoted by $\psi(x)$, to represent $f(x)$ with a few parameters. The basic rule for generating this basis is to translate and dilate $\psi(x)$, *i.e.*, $\psi(2^j x - i)$, where $j$ and $i$ are integers. The basis is localized in both time (controlled by $i$) and scale/frequency (controlled by $j$) domains. The original function $f(x)$ is decomposed as a linear combination of the basis:

$$f(x) = \sum_{j,i} a_{j,i} \psi(2^j x - i) \ , \tag{1}$$

where $a_{j,i}$ are the wavelet coefficients of $f(x)$ and can be computed as the inner product of $\psi(2^j x - i)$ with $f(x)$. For a discretized signal $x[n]$ ($n \in Z$), we sample points from $\psi(2^j x - i)$ to form filter banks to decompose $x[n]$. This is classical or first-generation wavelet (Santosa, 2011). Some commonly used mother wavelets include Haar wavelet (Haar, 1910), Daubechies wavelet, Biorthogonal wavelet (Daubechies, 1992), to name a few. Wavelet transform provides a nice framework for multi-resolution analysis of functions, *i.e.*, wavelet transform generates coarser and coarser versions of the same function while keeping track of the details missed going from fine to coarse. However, the first-generation wavelet mainly works well for infinite or periodic signals, which are rare in many applications. It is still unclear how to design a proper basis for any arbitrary signals sampled irregularly from an unbounded domain. One solution is to give up designing a mother wavelet manually. Instead, the decomposition is entirely based on an approach called *lifting scheme*.

**Lifting Scheme.** Lifting scheme, *a.k.a.*, second-generation wavelets (Sweldens, 1998; Daubechies & Sweldens, 1998), is a simple yet effective approach to define wavelets sharing the same properties as the first-generation wavelets. The lifting scheme takes as input a signal $x$ and generates as outputs the approximation $c$ and the detail $d$ sub-bands of the wavelet transform. The lifting scheme consists of three stages (Claypoole et al., 2003; Rodriguez et al., 2020) as on the left panel of Figure 2.

- *Split.* This stage splits an input signal into two non-overlapping partitions, *e.g.*, the input signal $x$ is divided into even $x_e$ and odd $x_o$ components, which are defined as $x_e[n] = x[2n]$ and $x_o[n] = x[2n + 1]$.

- *Update (U).* This stage separates the signal in frequency domain, generating the approximation $c$ that has the same running average as the original input signal. Let $x_o^{L_U}[n] = x_o[n - L_U], x_o[n - L_U + 1], \cdots, x_o[n + L_U - 1], x_o[n + L_U]$ denote the sequence of $2L_U + 1$ neighboring odd polyphase samples around $x_e[n]$. The even polyphase samples are updated through an update operator $U(\cdot)$ on $x_o^{L_U}[n]$, resulting the approximation $c$ as:

$$c[n] = x_e[n] + U(x_o^{L_U}[n]) \ . \tag{2}$$

- *Predict (P).* As the partitions $x_e$ and $x_o$ are usually closely correlated, given one of them, it is feasible to build a predictor $P(\cdot)$ for the other, by tracking the difference (or detail) $d$ among them. As the even component $x_e$ corresponds to the approximation $c[n]$, it is possible to define $P(\cdot)$ as a function of $c[n]$. Let $c^{L_P}[n] = c[n - L_P], c[n - L_P + 1], \cdots, c[n + L_P - 1], c[n + L_P]$ denote a sequence of $2L_P + 1$ approximation coefficients. The odd polyphase samples are predicted as $P(c^{L_P}[n])$. The resulting prediction residual, or detail $d$, are computed as:

$$d[n] = x_o[n] - P(c^{L_P}[n]) \ . \tag{3}$$

Note that the update and predict operators $U$ and $P$ implicitly and effectively define the wavelet for signal decomposition.

## 4 ADAPTIVE WAVELET TRANSFORMER NETWORK

**Overview.** In this section, we propose an Adaptive Wavelet Transformer Network, dubbed as AWT-Net, for 3D shape representation learning, as shown on the right panel of Figure 3. Similar to the overall structure of GDA-Net (Xu et al., 2021b), AWT-Net consists of three modules: 1) k-NN graph construction module to build graphs from point clouds, 2) adaptive lifting module, reflecting our main contribution that decomposes 3D shapes into high- and low-frequency components using wavelet analysis, and 3) transformer module to fuse decomposed component features. The AWT-Net generates shape representation hierarchically, and the decomposition tree is shown on the left panel of Figure 3. Here we show three scales where each scale corresponds to a separate lifting process as illustrated on the left panel of Figure 2. We omit the split, update and predict blocks for conciseness. At each scale, the approximation $c_i$ serves as a coarse version of the input signal while the detail $d_i$ maintains the detailed difference between $c_i$ and the input signal. The approximation $c_i$ is further decomposed by another lifting process at the next scale. The AWT-Net is built based on this decomposition tree but replaces the linear operators $U$ and $P$ in Eq. 2 and Eq. 3 with non-linear neural networks. Once trained and fixed, the non-linear update $U$ and predict $P$ operators can be regarded as two sub-band basis/filters to subdivide 3D shapes into high-frequency and low-frequency components, in analogy to the basis in Eq. 1.

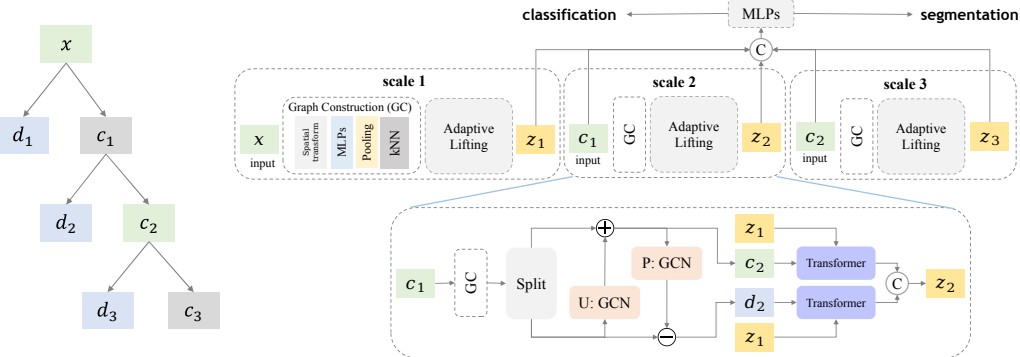

Figure 3: (Left) A multi-resolution wavelet decomposition tree (of three scales) with lifting scheme. (Right) An unrolled version of AWT-Net corresponding to the decomposition tree. Here we omit update and predict operators for simplicity.

### 4.1 EVEN-ODD SPLIT

**Graph Construction.** Assume a point cloud containing $N$ points associated with features $x_i \in \mathbb{R}^d, i \in \{1, \cdots, N\}$. Features can be coordinates, normals and/or other high-dimensional features. We construct a k-NN graph $\mathcal{G} = (\mathcal{V}, \tilde{\mathcal{A}}^u, \tilde{\mathcal{A}}^w)$, where each point $x_i$ corresponds to a node $v_i \in \mathcal{V}$, $\tilde{\mathcal{A}}^u$ and $\tilde{\mathcal{A}}^w \in \mathbb{R}^{N \times N}$ are unweighted and weighted adjacency matrices encoding point similarity in feature space. The unweighted and weighted edges connecting two nodes $v_i$ and $v_j$ are defined as:

$$\tilde{\mathcal{A}}^u_{ij} \ [, \tilde{\mathcal{A}}^w_{ij}] = \begin{cases} 1 \ [, \kappa(\|x_i - x_j\|_2)], & x_j \in \mathcal{N}(x_i) \\ 0, & \text{otherwise} \end{cases}, \tag{4}$$

where $\kappa(\cdot)$ is an non-negative function, *e.g.*, Gaussian function, to ensure that $\tilde{\mathcal{A}}^w$ is a diagonally dominant matrix and $\mathcal{N}$ represents neighborhood. Before constructing k-NN graph, we compute point features through a spatial transform layer (Wang et al., 2019), also termed as *local operator* in (Xu et al., 2021b). After graph construction, we normalize edge weights in each row in $\mathcal{A}^w$ to remove the impacts of varying neighborhood sizes and feature scales as in (Xu et al., 2021b).

**Graph Split.** We partition all nodes in a graph into two sets, containing *even* and *odd* nodes, respectively. Since lifting scheme uses information from odd (*resp.* even) nodes to update (*resp.* predict) even (*resp.* odd) nodes as in Eq. 2 (*resp.* Eq. 3), having all neighboring nodes with same parity means that local information cannot be used. Following (Narang & Ortega, 2009), we define the number of

---

**Algorithm 1:** Even-Odd Node Split

---

**Require:** A graph $\mathcal{G} = (\mathcal{V}, \mathcal{A}^u)$ with unweighted adjacency, maximum number of iterations $M$

1 Randomly assign initial parity (*e.g.*, 1 for odd, 0 for even) to each node
2 **for** *m = 1:M* **do**
3      Activate each node with a probability sampled from a uniform distribution independently
4      **for** *each activated node* **do**
5          Choose a parity that minimizes its conflict with neighboring nodes
     `/* Keep the number of odd and even nodes roughly the same     */`
6      $N_o, N_e \leftarrow$ number of nodes with parity 1, parity 0
7      **if** $N_o > N_e$ **then**
8          Randomly flip $\lfloor \frac{N_o - N_e}{2} \rfloor$ odd nodes' parity
9      **else**
10          Randomly flip $\lfloor \frac{N_e - N_o}{2} \rfloor$ even nodes' parity

---

conflicts as the percentage of neighboring nodes that have the same parity after partition. The optimal splitting minimizes the number of conflicts. Theoretically, it is an NP-hard bipartite sub-graph problem, but a heuristic solution can be found. We adapt the conservative fixed probability colorer algorithm in (Fitzpatrick & Meertens, 2001), which is based on iteratively greedy local heuristics: at each iteration, a few randomly chosen nodes are activated, and each activated node counts the number of conflicting edges (*i.e.*, connecting neighboring nodes with the same parity) and changes its parity based on the conflict. The algorithm is summarized in Algorithm 1.

### 4.2 ADAPTIVE LIFTING SCHEME

Based on nodes' parity after even-odd split, we re-arrange nodes $\mathcal{V}$ to gather odd and even nodes, and also re-arrange the weighted adjacency matrix $\mathcal{A}^w$ accordingly:

$$\mathcal{V} = \begin{pmatrix} \mathcal{V}_{odd} \\ \mathcal{V}_{even} \end{pmatrix} \quad , \quad \mathcal{A}^w = \begin{pmatrix} \mathcal{F}_{N_o \times N_o} & \mathcal{J}_{N_o \times N_e} \\ \mathcal{K}_{N_e \times N_o} & \mathcal{L}_{N_e \times N_e} \end{pmatrix} \quad ,$$

where $\mathcal{V}_{odd}$ and $\mathcal{V}_{even}$ represent the sets of odd and even nodes. The sub-matrix $\mathcal{F}$ is the adjacency matrix of a subgraph containing odd nodes only. The submatrix $\mathcal{J}$ contains edges that do not have conflicts, *i.e.*, edges connecting nodes with the opposite parity. Similar definitions for $\mathcal{K}$ and $\mathcal{L}$.

Unlike the linear lifting scheme introduced in Section 3, we propose data-dependent non-linear update and predict operators, implemented as neural networks whose parameters are adaptively optimized to capture complex shape geometry. Specifically, we apply two Graph Convolutional Networks (GCNs) proposed in (Kipf & Welling, 2017) as the update and predict operators, separately. The graph convolution allows us to compute the response of a node based on its neighbors decided by the adjacency matrix. Performing graph convolution is equal to performing message gathering within a neighborhood around each center node, sharing similarities with the definitions of $x_o^{L_u}[n]$ and $c^{L_p}[n]$ in Eq. 2 and Eq. 3. The non-linear update and predict stages at $i$-th scale are defined as:

$$c_i = \mathcal{V}_{even} + \text{GCN}(\mathcal{V}_{odd}, \mathcal{F}, \mathcal{K}) \quad , \tag{5}$$

$$d_i = \mathcal{V}_{odd} - \text{GCN}(c_i, \mathcal{L}, \mathcal{J}) \quad , \tag{6}$$

where $c_i$ and $d_i$ represent the approximation and detail components at $i$-th scale. We repeat this process at each scale recursively and form a hierarchical decomposition tree, capturing shape geometry at multi-scales. The maximum number of scales can be decided analytically as $\log_2 N$ and the coarsest scale contains only a single point.

### 4.3 TRANSFORMER FOR CROSS ATTENTION

The approximation and detail components depict 3D shape geometry from different but complementary aspects, and thus can be exploited to enhance the original input shape features. Similar to GDA-Net (Xu et al., 2021b) which adopts an attention module to fuse variation components with the original shape features, we adopt Transformer (Vaswani et al., 2017) to query and fuse features from different sub-bands gained from lifting scheme in a principled way. The architecture of Transformer is shown on the right panel of Figure 2, and multi-head attention is defined as:

$$\text{Attention}(Q, K, V) = \text{softmax}(QK^\top)V \quad , \tag{7}$$

where $Q, K, V$ denote query, key and value, respectively.

For approximation $c_i$ as coarse component at $i$-th scale, we set query, key and value to:

$$Q = z_{i-1}, K = c_i, V = c_i \ , \tag{8}$$

where the definition of $z_i$ is shown in Figure 3. Note that for the first scale (*i.e.*, $i = 1$), we set $Q = z_{i-1} = x$ as input. For detail $d_i$, as most of the values are close to zeros (explained in Section 4.4), directly feeding them into Transformer yields sub-optimal results. Utilizing a nice property of wavelets, namely time-frequency localization[1], we regard $d_i$ as a multiplier to strengthen or weaken the context features. Formally, the query, key and value for detail $d_i$ at $i$-th scale are set as:

$$Q = z_{i-1}, K = c_{i-1} \otimes \exp(|d_i|), V = c_{i-1} \otimes \exp(|d_i|) \ , \tag{9}$$

where $\otimes$ represents element-wise multiplication. For the first scale, we set both $z_{i-1}$ and $c_{i-1}$ to $x$.

### 4.4 LOSS FUNCTION

End-to-end training is performed using task-specific cross-entropy loss, in combination with regularization terms, to enforce a wavelet decomposition structure during training. The loss function takes the form of Eq. 10, where $K$ in the first term denotes the number of classes, $y_i$ and $p_i$ are the ground-truth and the predicted probability of belonging to class $i$, respectively.

$$\text{Loss} = -\sum_{i=1}^{K} y_i \log(p_i) + \lambda_1 \sum_{i=1}^{S} \text{Smooth}_{L1}(\mathcal{A}c_{i-1} - c_i) + \lambda_2 \sum_{i=1}^{S} \text{Smooth}_{L1}(d_i) \ . \tag{10}$$

In lifting scheme, a perfect output of Eq. 2, *i.e.*, the series $c[n]$, would be a coarse approximation of $x[n]$ at half the resolution. In the second term, we enforce each lifted even node has the same local average as its corresponding node in the original graph. Due to the half resolution of $c_i$ compared with the input $c_{i-1}$, we multiply the adjacent matrix $\mathcal{A}$ with $c_{i-1}$ to aggregate the node features within the neighborhood, and minimize the difference between the input graph features and the coarser version across all the decomposition scales. The rationale for the third term is: if the underlying signal $x[n]$ is locally smooth, the prediction residuals $d[n]$ should be small. Thus, the third term promotes the minimization of the magnitude of detail coefficients $d[n]$ in Eq. 3, indicating smaller energy is contained in $d[n]$.

## 5 EXPERIMENTS AND RESULTS

We evaluate our model on 3D shape classification and part segmentation tasks on three datasets following the settings (Qi et al., 2017a;b; Wang et al., 2019; Xu et al., 2021b; Zhao et al., 2021): ModelNet40 (Wu et al., 2015), ScanObjectNN (Uy et al., 2019) and ShapeNet Part (Yi et al., 2016).

### 5.1 IMPLEMENTATION DETAILS

The number of the constructed neighborhoods in k-NN graphs at each stage is 64. The maximum number of iterations $M$ used in the split Algorithm 1 is set to 10, which strikes a balance between efficiency and accuracy. The update $U(\cdot)$ and $P(\cdot)$ operators are implemented as two GCNs (Kipf & Welling, 2017) and the internal structure of each GCN is: Conv-BN-ReLU-Dropout-Conv-BN. We use a two-heads attention in Transformer. We implement our model in PyTorch (Paszke et al., 2019). We use SGD optimizer with momentum and weight decay set to 0.9 and 0.0001, respectively. For both 3D shape classification and 3D object part segmentation tasks, we train for 350 epochs. The initial learning rate is set to 0.2 for shape classification and 0.005 for object part segmentation, and clipped at $0.9e - 5$. The values of $\lambda_1$ and $\lambda_2$ are set to 0.1 for all tasks.

### 5.2 OBJECT CLASSIFICATION

We evaluate AWT-Net on ModelNet40 (Wu et al., 2015) which contains 9,843 training shapes and 2,468 test shapes across 40 categories. Points are uniformly sampled from mesh models by (Qi et al., 2017a). Following (Wang et al., 2019; Xu et al., 2021b), we translate shapes and randomly shuffle points in the training set. Similar to (Qi et al., 2017a;b), we conduct several voting tests with random scaling and average the predictions during testing. The quantitative comparisons of the

---

[1]In analogy wave signals expanding time domain, for 3D shapes which expand in the spatial domain, this property can be rephrased as spatial-frequency localization.

overall classification accuracy with the state-of-the-art methods are in Table 1. AWT-Net with two scales achieves the best performance. Beyond quantitative evaluation, we exhibit that our method can decompose shapes into approximation and detail components as shown in Figure 1 and Figure 6.

We also validate the robustness of AWT-Net on a real-scanned dataset (Uy et al., 2019) following the experimental settings in GDA-Net (Xu et al., 2021b). We test on the OBJ_ONLY and noisy OBJ_BG splits, and the results are summarized in Table 2, where our model achieves comparable accuracy and performance drop from OBJ_ONLY to OBJ_BG.

| Method (time order) | Points | Accuracy |
|---|---|---|
| PointNet (Qi et al., 2017a) | 1K | 89.2 |
| PointNet++ (Qi et al., 2017b) | 1K | 90.7 |
| SCN (Xie et al., 2018) | 1K | 90.0 |
| KCNet (Shen et al., 2018) | 1K | 91.0 |
| PointCNN (Li et al., 2018b) | 1K | 92.2 |
| Point2Sequence (Liu et al., 2019a) | 1K | 92.6 |
| DGCNN (Wang et al., 2019) | 1K | 92.9 |
| InterpCNN (Mao et al., 2019) | 1K | 93.0 |
| RS-CNN (Liu et al., 2019b) | 1K | 93.6 |
| 3D-GCN (Lin et al., 2020b) | 1K | 92.1 |
| FPConv (Lin et al., 2020a) | 1K | 92.5 |
| GSNet (Xu et al., 2020) | 1K | 92.9 |
| PointTransformer (Zhao et al., 2021) | 1K | 93.7 |
| PCT (Guo et al., 2021) | 1K | 93.2 |
| GDA-Net (Xu et al., 2021b) | 1K | 93.8 |
| **AWT-Net (ours)** | **1K** | **93.9** |

Table 1: Classification accuracy (%) on ModelNet40 dataset.

| Method | OBJ_ONLY | OBJ_BG | Acc. drop |
|---|---|---|---|
| PointNet (Qi et al., 2017a) | 79.2 | 73.3 | ↓ 5.9 |
| PointNet++ (Qi et al., 2017b) | 84.3 | 82.3 | ↓ 2.0 |
| SpiderCNN (Xu et al., 2018) | 79.5 | 77.1 | ↓ 5.4 |
| PointCNN (Li et al., 2018b) | 87.9 | 85.8 | ↓ 2.1 |
| DGCNN (Wang et al., 2019) | 86.2 | 82.8 | ↓ 3.4 |
| GDA-Net (Xu et al., 2021b) | **88.5** | **87.0** | ↓ 1.5 |
| **AWT-Net (ours)** | **88.5** | 86.8 | ↓ 1.7 |

Table 2: Classification accuracy (%) *w.r.t.* noise on ScanObjectNN dataset.

## 5.3 PART SEGMENTATION

We evaluate AWT-Net with three scales for shape part segmentation on ShapeNet Part (Yi et al., 2016) dataset, which contains 16,881 shapes in 16 categories and is labeled in 50 parts where each shape has 2~5 parts. The same voting test in classification is adopted. The instance average, the class average, and each class mean Inter-over-Union (mIoU) are summarized in Table 3. AWT-Net achieves the best performance with class mIoU of 85.0% and instance mIoU of 86.6%. Some qualitative segmentation results are visualized in Figure 4.

| Method (time order) | class mIOU | instance mIOU | aero | bag | cap | car | chair | ear phone | guitar | knife | lamp | lap top | motor | mug | pistol | rocket | skate board | table |
|---|---|---|---|---|---|---|---|---|---|---|---|---|---|---|---|---|---|---|
| PointNet (Qi et al., 2017a) | 80.4 | 83.7 | 83.4 | 78.7 | 82.5 | 74.9 | 89.6 | 73.0 | 91.5 | 85.9 | 80.8 | 95.3 | 65.2 | 93.0 | 81.2 | 57.9 | 72.8 | 80.6 |
| PointNet++ (Qi et al., 2017b) | 81.9 | 85.1 | 82.4 | 79.0 | 87.7 | 77.3 | 90.8 | 71.8 | 91.0 | 85.9 | 83.7 | 95.3 | 71.6 | 94.1 | 81.3 | 58.7 | 76.4 | 82.6 |
| SyncCNN (Yi et al., 2017) | 82.0 | 84.7 | 81.6 | 81.7 | 81.9 | 75.2 | 90.2 | 74.9 | **93.0** | 86.1 | 84.7 | 95.6 | 66.7 | 92.7 | 81.6 | 60.6 | 82.9 | 82.1 |
| SCN (Xie et al., 2018) | 81.8 | 84.6 | 83.8 | 80.8 | 83.5 | 79.3 | 90.5 | 69.8 | 91.7 | 86.5 | 82.9 | 96.0 | 69.2 | 93.8 | 82.5 | 62.9 | 74.4 | 80.8 |
| KCNet (Shen et al., 2018) | 82.2 | 84.7 | 82.8 | 81.5 | 86.4 | 77.6 | 90.3 | 76.8 | 91.0 | 87.0 | 84.5 | 95.5 | 69.2 | 94.4 | 81.6 | 60.1 | 75.2 | 81.3 |
| SpiderCNN (Xu et al., 2018) | 82.4 | 85.3 | 83.5 | 81.0 | 87.2 | 77.5 | 90.7 | 76.8 | 91.1 | 87.3 | 83.3 | 95.8 | 70.2 | 93.5 | 82.7 | 59.7 | 75.8 | 82.8 |
| Point2Sequence (Liu et al., 2019a) | - | 85.2 | 82.6 | 81.8 | 87.5 | 77.3 | 90.8 | 77.1 | 91.1 | 86.9 | 83.9 | 95.7 | 70.8 | 94.6 | 79.3 | 58.1 | 75.2 | 82.8 |
| DGCNN (Wang et al., 2019) | 82.3 | 85.2 | 84.0 | 83.4 | 86.7 | 77.8 | 90.6 | 74.7 | 91.2 | 87.5 | 82.8 | 95.7 | 66.3 | 94.9 | 81.1 | 63.5 | 74.5 | 82.6 |
| RS-CNN (Liu et al., 2019b) | 84.0 | 86.2 | 83.5 | 84.8 | 88.8 | 79.6 | 91.2 | 81.1 | 91.6 | 88.4 | 86.0 | 96.0 | 73.7 | 94.1 | 83.4 | 60.5 | 77.7 | 83.6 |
| 3D-GCN (Lin et al., 2020b) | 82.1 | 85.1 | 83.1 | 84.0 | 86.6 | 77.5 | 90.3 | 74.1 | 90.0 | 86.4 | 83.8 | 95.6 | 66.8 | 94.8 | 81.3 | 59.6 | 75.7 | 82.8 |
| GSNet (Xu et al., 2020) | 83.5 | 85.3 | 82.9 | 84.3 | 88.6 | 78.4 | 89.7 | 78.3 | 91.7 | 86.7 | 81.2 | 95.6 | 72.8 | 94.7 | 83.1 | 62.3 | 81.5 | 83.8 |
| PCT (Guo et al., 2021) | - | 86.4 | **85.0** | 82.4 | 89.0 | **81.2** | **91.9** | 71.5 | 91.3 | 88.1 | **86.3** | 95.8 | 64.6 | 95.8 | 83.6 | 62.2 | 77.6 | 83.7 |
| GDA-Net (Xu et al., 2021b) | **85.0** | 86.5 | 84.2 | 88.0 | **90.6** | 80.2 | 90.7 | **82.0** | 91.9 | 88.5 | 82.7 | 96.1 | **75.8** | 95.7 | **83.9** | 62.9 | 83.1 | **84.4** |
| PointTransformer (Zhao et al., 2021) | 83.7 | **86.6** | - | - | - | - | - | - | - | - | - | - | - | - | - | - | - | - |
| **AWT-Net (ours)** | **85.0** | **86.6** | 83.9 | **88.4** | 88.5 | 79.2 | 90.6 | 81.3 | 92.5 | **88.7** | 82.1 | **96.3** | 73.9 | **95.9** | 83.5 | **63.0** | **84.1** | 84.2 |

Table 3: Segmentation results (%) on ShapeNet Part dataset.

## 6 ABLATION STUDY

We conduct ablative experiments on ModelNet40 and no voting during testing (unless specified). In all ablative experiments, we follow the same experimental settings DGA-Net (Xu et al., 2021b). In addition, we adopt the same performance measurement metrics that were used in DGA-Net to report our results in Table 4 to Table 8 and Figures 5 in validating and analyzing the effectiveness of our 3D shape representation learning approach based on wavelet analysis for shape decomposition.

**Module Analysis.** We ablate each module in AWT-Net, and the results are listed in Table 4. When the Transformer takes as input the original point cloud features fused with both approximation and detail components, AWT-Net achieves the best accuracy with 93.4%. Our method ultimately obtains an accuracy of 93.9% with voting. This experiment shows that the decomposed approximation and detail component complement mutually capture different but complementary geometric features to get complete shape representation.

**Split Scheme.** We validate the split of approximation and detail components against random and farthest point sampling (FPS). The results are listed in Table 5, where decomposing point clouds into approximation and detail components gets the highest accuracy.

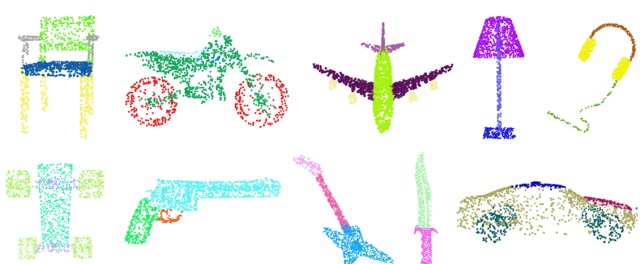

**Adjacency Matrix Computation.** The result of dynamic adjacency matrix computation in Section 4.1 is shown in Table 6. We pre-compute the adjacency matrix on 3D coordinates to split point clouds, which achieves the accuracy of 91.8%. Nevertheless, constructing the adjacency matrix in high dimensional semantic space at different scales gains 1.6% improvement. Figure 1 and Figure 6 show that our decomposition strategy successfully decomposes points into flat areas and areas containing edges. Note that we do not provide any supervision *w.r.t.* shape geometry.

Figure 4: Qualitative results of part semantic segmentation.

**Decomposition Levels.** We investigate the impact of decomposition levels (*i.e.*, resolution scales), and the results are summarized in Table 7. AWT-Net performs best on the overall accuracy when the decomposition level is set to 2. We also observe that for shapes with simple geometric structures (*e.g.*, radio and wardrobe), more scales have a negative effect on the performance, as more scales make the network prone to over-fitting on these simple shapes. While for shapes with complex structures (*e.g.*, stool and person), more scales are needed to capture both coarse and fine-grained geometric features. That is, the accuracy increases with more scales applied to geometrically more complex objects. We find out that the ModelNet40 dataset contains more geometrically simple objects than complex ones, which leads to the overall accuracy drops with more scales adopted.

**Sampling and Rotation Test.** First, we test AWT-Net on the 3D shapes with different numbers of randomly sampled points and Figure 5 lists the results. Next, we evaluate the robustness of AWT-Net against rotation and Table 8 summarizes the results. We note that AWT-Net outperforms the compared methods.

## 7 CONCLUSION

Previous works such as GDA-Net have observed the effectiveness of learning shape representation via decomposing the 3D shapes into low- and high-frequency components. In this work, we further advance the shape decomposition with a novel wavelet analysis based method that decomposes 3D shapes into sub-bands components. Our work proposed to use the deep neural networks to realize the shape decomposition in a learnable fashion that seamlessly connects to the downstream tasks of shape representation learning. Extensive experiments show that our AWT-Net achieves excellent performance on 3D shape classification and segmentation tasks which are on par with the state-of-the-art approaches.

ACKNOWLEDGMENTS

We would like to thank the reviewers and the authors of GDA-Net (Xu et al., 2021b) for their comments and efforts towards improving our manuscript. The authors appreciate the generous support provided by Inception Institute of Artificial Intelligence (IIAI) in the form of NYUAD Global Ph.D. Student Fellowship. This work was also partially supported by the NYUAD Center for Artificial Intelligence and Robotics (CAIR), funded by Tamkeen under the NYUAD Research Institute Award CG010.

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

# A APPENDIX

## A.1 TABLES AND FIGURES FOR ABLATION STUDY

In this section, we list tables and figures for ablation study in Section 6 for a better understanding of the proposed model.

| No. | k-NN | self | approximation | detail | voting | accuracy (%) |
|---|---|---|---|---|---|---|
| 1 | | | | | | 91.4 |
| 2 | ✓ | | | | | 92.3 |
| 3 | | ✓ | | | | 89.8 |
| 4 | ✓ | ✓ | | | | 92.4 |
| 5 | | | ✓ | ✓ | | 92.9 |
| 6 | ✓ | | ✓ | | | 92.5 |
| 7 | ✓ | | | ✓ | | 92.7 |
| 8 | ✓ | | ✓ | ✓ | | **93.4** |
| 9 | ✓ | | ✓ | ✓ | ✓ | **93.9** |

Table 4: Results of ablating each module in AWT-Net on ModelNet40.

| Method | ModelNet40 | OBJ_ONLY | OBJ_BG |
|---|---|---|---|
| random | 89.6 | 82.9 | 83.8 |
| FPS | 92.0 | 85.3 | 84.2 |
| app.-det. | 93.4 | 88.0 | 86.3 |

Table 5: Effect of different point split schemes on shape classification.

| Method | Accuracy |
|---|---|
| Pre-computed | 91.8 |
| Dynamic | 93.4 |

Table 6: Effect of adjacency matrix computation schemes.

| Number of scales | 1 | 2 | 3 | 4 |
|---|---|---|---|---|
| Overall | 92.5 | 93.4 | 93.1 | 91.3 |
| Radio | 90.0 | 83.5 | 75.0 | 75.0 |
| Wardrobe | 75.0 | 71.0 | 68.0 | 65.0 |
| Stool | 75.0 | 78.0 | 85.0 | 85.2 |
| Person | 85.0 | 88.5 | 95.0 | 100.0 |

Table 7: Effect of decomposition levels on ModelNet40. Note that the results are achieved without voting during test.

| Method | z-rot | xyz-rot |
|---|---|---|
| PointNet (Qi et al., 2017a) | 81.6 | 66.3 |
| PointNet++ (Qi et al., 2017b) | 90.1 | 87.8 |
| SpiderCNN (Xu et al., 2018) | 83.5 | 69.6 |
| DGCNN (Wang et al., 2019) | 90.4 | 82.6 |
| GDA-Net (Xu et al., 2021b) | 91.2 | 90.5 |
| AWT-Net (ours) | 91.4 | 90.2 |

Table 8: Accuracy on ModelNet40 against rotation. z-rot: randomly rotate training and test shapes along z axis; xyz-rot: randomly rotate training and test shapes along xyz axis.

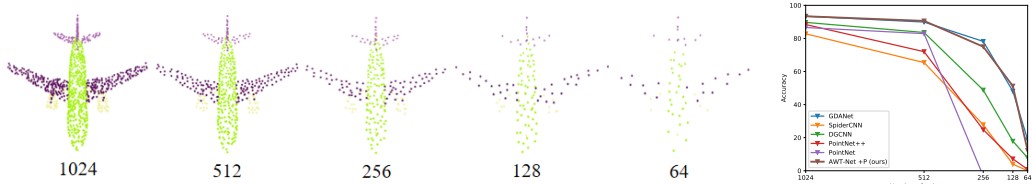

Figure 5: On the right, the curves show the classification results of AWT-Net tested on the 3D shapes with different numbers of randomly sampled points (shown on left).

## A.2 MORE VISUALIZATION OF DECOMPOSITION OF SHAPES

We present more visualization of detail components of multiple shapes from the test set of Model-Net40 (Wu et al., 2015) in Figure 6. Such geometric representations are learned implicitly in 3D shape classification task without component supervision.

## A.3 EXECUTION TIME AND MODEL PARAMETERS

We compare the number of parameters and computational overhead *w.r.t.* GDA-Net (Xu et al., 2021b) on ModelNet40 and the results are listed in Table 9. We measure four scales. *Note that GDA-Net consists of two building blocks, which is fairly comparable to "scale 2" in our model.* In this case, though our model contains around 25% more parameters, the execution time is only 13% slower than GDA-Net, which shows the practicability of our model.

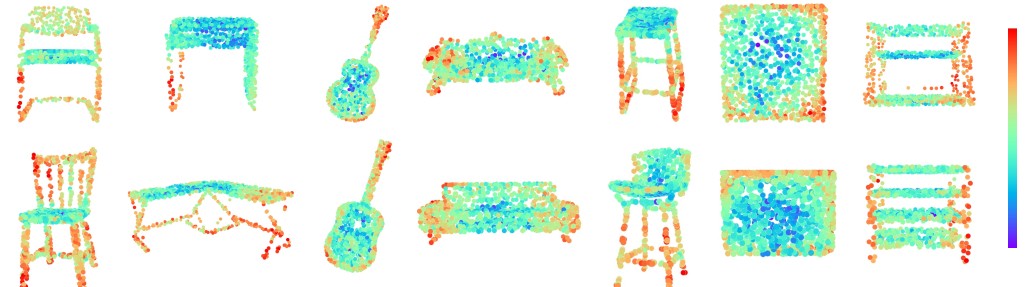

Figure 6: The detail component of chair, table, guitar, sofa, stool, wardrobe, and shelf. Colors represent the absolute values of the detail coefficients per point. The larger values (redder colors), the higher probability that these points locate on high-frequency parts, *e.g.*, edges or non-flat areas.

| Model | Number of Params | Time |
|---|---|---|
| GDA-Net (Xu et al., 2021b) | 0.94 M | 1.057 |
| AWT-Net (scales 1) | 0.97 M | 0.843 |
| **AWT-Net (scales 2)** | **1.17 M** | **1.191** |
| AWT-Net (scales 3) | 1.38 M | 1.657 |
| AWT-Net (scales 4) | 1.60 M | 2.024 |

Table 9: The number of parameters and computational overhead (time) comparison. The computational overhead is measured in second for a single forward and backward pass. All results are average over ten trails.

## A.4 RANDOMNESS IN EVEN-ODD NODE SPLIT

We compare the effect of using even-odd split and random split in Algorithm 1 and the results are listed in Table 10. Using random split performers slightly worse than even-odd split. With even-odd split, we can separate any two connected nodes into two different sets to the best extent, thus ensuring each local area contains both even and odd nodes. However, with random split, some local areas might contain only one type of nodes, thus failing to be decomposed properly.

| Number of scales | 1 | 2 | 3 | 4 |
|---|---|---|---|---|
| Even-odd split | 92.5 | 93.4 | 93.1 | 91.3 |
| Random split | 91.7 | 92.9 | 92.6 | 90.9 |

Table 10: Classification results of using even-odd split and random split in Algorithm 1.

