# OpenReview forum: "Adaptive Wavelet Transformer Network for 3D Shape Representation Learning"
_ICLR.cc/2022/Conference — ICLR 2022 Poster_

### Official Review · Reviewer_tf1v · 2021-11-01

**Correctness:** 3
**Technical Novelty And Significance:** 3
**Empirical Novelty And Significance:** 2
**Recommendation:** 6
**Confidence:** 4

**Main Review:**

Pros:
1. The paper is well written and provides sufficient background knowledge.
2. Applying wavelet transform for 3D representation learning seems quite new and is not much explored before.
3. For the experiment results, the numbers themselves are stronger and ablation study is well conducted.

Cons & questions:
1. My biggest concern is about the results comparison, even though the final number itself is strong. Table 1 shows the proposed AWT-Net achieves the best results when using Performer (AWT-Net +P). It a little bit unfair since others transformer-based prior works (e.g. PointTransformer, PCT) use vanilla Transformer. Their results might also be improved with Performer. If we look at only results for 'AWT-Net +V' in Table 1 and Table 3, the numbers fall behind prior transformer-based works like PointTransformer and PCT. Is using Performer a must? In addition, why doesn't Table 2 also show the result for 'AWT-Net + V'? I think it's better to put it on if there are no particular reasons.
2. The module analysis of ablation study is sort of unclear to me. First, what exactly is the baseline (the first row in table 4)? It's unclear to me what's the network architecture after ablating all the modules. I feel it's better to have a supplementary document describing the details. Second, the improvement (from 92.3 to 93.4) by adding ‘approximation’ and ‘detail’ doesn't seem that much. It is because this task is already saturated? Or gives a convincing evidence that it is a considerable improvement.
Is there any other way to demonstrate the effeteness of using wavelet decomposition? The visualization in Figure 1 is a good example to understand the learned features but could potentially be explored more (e.g. how is hierarchy/multi-resolution reflected here).
3. I'm also curious about the execution time. The proposed method is quite complex (with split scheme, GCN and Transformer), so I wonder if it will be much slower than previous works. A table showing the execution time and model parameters would be much helpful.
4. Will the randomness in even-odd node split (Algorithm 1) affect the results? Just want to confirm different node partition for the same 3D shape does not much affect the learned feature.

Minor point:
- The resolution of Figure 5 right is too low. Consider using a pdf version.

**Summary Of The Paper:**

The paper presents a new deep neural network architecture for 3D point cloud representation learning, based on wavelet decomposition. In particular, the authors propose a data-driven adaptive lifting scheme that introduces non-linearity into wavelet. The original linear operators update(U) and predict(P) in wavelet decomposition are replaced by non-linear graph convolutional networks (GCN). Equipped with wavelet transform and Transformers, the proposed network aims to captures and refines the holistic and complementary geometry of 3D shapes to supplement neighboring local information. Experimental results on standard benchmarks (i.e., shape classification and part segmentation) show that it achieves state-of- the-arts or competitive performance.

**Summary Of The Review:**

In general, the proposed method is novel in that it combines wavelet transform into 3D representation learning (I'm not an expert in wavelet transform, so I'm not quite sure how challenging this is). Experiments and ablation study are comprehensive, and the final quantitative numbers are competitive. My biggest concern is that the improvement over prior works are quite limited (no more than 0.2%), given all the complex schemes of the method and its required usage of Performer. I understand these benchmark results are somewhat saturated and hard to improve a lot. It's ok, but the paper will be more convincing if the authors can analyze the features learned by their wavelet transform, and show how the learned features are better or different from prior works that do not use wavelet transform.

---

> ### Author Response · Authors · 2021-11-23
> **Response to Reviewer tf1v - Part 1**
>
> **Q1**. Concern about the results comparison.
> **A1**. The performance of our model is compared with prior methods on the two widely used but saturated benchmarks. Thus, we sincerely appreciate you can be more tolerant to the marginal performance improvement or difference (i.e. 0.1%, 0.3%). Using Performer is not a must. Our initial motivation is to use Performer to speed up the training process by reducing the quadratic time complexity of attention used in the vanilla Transformer, not intended to improve the performance. As shown in the new Table 11 in Appendix, Performer is faster than the vanilla Transformer.
>
> In our opinion, the comparisons are fair, as the vanilla Transformer (AWT-Net +V (ours)) achieves the performance on par with the state-of-the-art methods, which is able to prove the effectiveness of our method. The PCT adopts an Offset-Attention as a variant of Transformer, rather than the purely vanilla one. For PointTransformer by Zhao et al., the proposed transformer layer has a different formulation (Eq.(3) in the paper) with the vanilla Transformer, which might not be a trivial job to be adapted to Performer. The comparisons are fair, as the vanilla Transformer (AWT-Net +V (ours)) achieves the performance on par with the state-of-the-art methods, which is able to prove the effectiveness of our method.
>
> Our initial motivation is to use Performer to speed up the training process by reducing the quadratic time complexity of attention used in the vanilla Transformer, not intended to improve the performance. As shown in the new Table 11 in Appendix, Performer is faster than the vanilla Transformer.  Please kindly refer to **A1** and **A2** in **General Responses to All Reviewers** section for more discussion regarding our motivation and experiments. If you recommend only keeping the vanilla Transofmer and removing Performer in order to make the comparison clearer to highlight the contributions of our wavelet transformer for 3D shape representation learning, we will respect the recommendation to remove Performer to avoid confusion and make it better highlight our contributions.
>
> The results for ’AWT-Net + V’ for ObjectScannNN are added to Table 2 in the revised manuscript.
>
> **Q2**. The module analysis of ablation study is sort of unclear to me. First, what exactly is the baseline (the first row in table 4)? It's unclear to me what's the network architecture after ablating all the modules. I feel it's better to have a supplementary document describing the details. Second, the improvement (from 92.3 to 93.4) by adding ‘approximation’ and ‘detail’ doesn't seem that much. It is because this task is already saturated? Or gives a convincing evidence that it is a considerable improvement.
>
> **A2**. Thanks for the constructive comments. Table 11 in the appendix is provided describing the configurations of the ablation study and Figure 3 is also updated to illustrate the configurations. Classifying 3D shapes based on the k-NN graphs constructed from point clouds has already achieved relatively high accuracy, eg, 92.9\% in DGCNN. Thus, we do not expect disentangling 3D shapes into high and low frequency components can bring significant performance improvement on these two almost saturated datasets.

---

> ### Author Response · Authors · 2021-11-23
> **Response to Reviewer tf1v - Part 2**
>
> **Q3**. Is there any other way to demonstrate the effeteness of using wavelet decomposition? The visualization in Figure 1 is a good example to understand the learned features but could potentially be explored more (e.g. how is hierarchy/multi-resolution reflected here).
>
> **A3**. In the revised manuscript, we have a more intuitive presentation to highlight our contributions. For example, in Figure 1, we added our observations that our model can learn meaningfully samplings of point clouds in different categories. And can detect key regions consistently within a semantic class. In Table 7, we find our multi-scale analysis is consistent with the geometric complexity of 3D shapes.
>
> **Q4**. I'm also curious about the execution time. The proposed method is quite complex (with split scheme, GCN and Transformer), so I wonder if it will be much slower than previous works. A table showing the execution time and model parameters would be much helpful.
>
> **A4**. The number of network parameters and computational overhead are added to the new Table 9 in the Appendix.
>
> **Q5**. Will the randomness in even-odd node split (Algorithm 1) affect the results? Just want to confirm different node partition for the same 3D shape does not much affect the learned feature.
>
> **A5**. The results of this experiment are added to the new Table 10 in the Appendix.
>
> **Q6**. My biggest concern is that the improvement over prior works are quite limited (no more than 0.2\%), given all the complex schemes of the method and its required usage of Performer. I understand these benchmark results are somewhat saturated and hard to improve a lot. It's ok, but the paper will be more convincing if the authors can analyze the features learned by their wavelet transform, and show how the learned features are better or different from prior works that do not use wavelet transform.
>
> **A6**. Please kindly refer to **A1**.
>
> **Q7**. Minor - The resolution of Figure 5 right is too low. Consider using a pdf version.
>
> **A7**. We have replaced Figure 5 right with a high-resolution pdf version.

---

### Official Review · Reviewer_Dnca · 2021-11-02

**Correctness:** 4
**Technical Novelty And Significance:** 4
**Empirical Novelty And Significance:** 3
**Recommendation:** 6
**Confidence:** 3

**Main Review:**

The strengths:
1) The paper is well-organized, the experiments are sufficient demonstrating the effectiveness of the proposed framework design.
2) Relying on wavelet decomposition for 3D shape learning is theoretically sound and interesting.
3) The authors made many efforts on the framework design, like the proposed new adaptive lifting scheme, the transformer etc.

The weaknesses:
My major concern is about the performance. It seems the performance is just comparable with existing methods. This brings a doubt to me: is it really necessary to use wavenet, a relatively new framework?

**Summary Of The Paper:**

This paper presents a novel framework for 3D shape representation learning, which is based on multi-scale wavelet decomposition. This is very different from existing works. A novel transformer-based neural network, AWT-Net, is also proposed.

**Summary Of The Review:**

Although the proposed method is theoretically sound and the framework design makes sense, the performance gain is hard to convince me the superiority of such a different learning framework.

---

> ### Author Response · Authors · 2021-11-23
> **Response to Reviewer Dnca**
>
> We appreciate the time and efforts you have spent to deeply understand the key contributions of our paper and provide valuable and positive comments. We have addressed your concerns by answering the following question.
>
> **Q1**. My major concern is about the performance. It seems the performance is just comparable with existing methods. This brings a doubt to me: is it really necessary to use wavenet, a relatively new framework?
>
> **A1**. The performance of our method is on par with the state-of-the-art methods on the two widely used but saturated benchmarks, which, in our opinion, is able to prove the effectiveness of our newly proposed adaptive wavelet transformer. Thus, we appreciate the reviewers could put more weight on our motivation and novel contribution of introducing wavelet transformer that uses sub-bands components for 3D shape representation learning, and be more tolerant to the marginal performance improvement or difference (i.e. 0.1%, 0.3%) on saturated benchmarks (e.g., ModelNet40).
>
> In general, our proposed method is motivated by the following observation. 1. The geometric components with different frequencies in 3D shapes contain distinct geometric characteristics (e.g., high frequency components correspond to edges / sharp parts, while low frequency components correspond to flat areas / smooth parts) and provide complementary geometric information to the shape representations. To effectively learn 3D shape representations, it is desirable to separate these components and process them differently, rather than feed the whole shapes into a single processor. It, therefore, motivates us to decompose 3D shapes into sub-bands components. 2. Wavelet analysis in the signal processing field provides a nice framework to decompose signals into sub-bands and thus we design our method based on wavelet analysis. However, the conventional lifting scheme in wavelet analysis adopts hand-crafted linear functions and cannot automatically adapt to data. Therefore, they are only suitable to a limited range of patterns in signals and cannot dynamically adapt to various patterns existing in a collection of point clouds. 3. These observations motivate us to propose an adaptive lifting scheme, a data-driven approach to automatically learn sub-band filter parameters to decompose 3D shapes, which makes our method learnable and be adaptive to different data and tasks. 4. On the two commonly used benchmarks, the performance of our method is comparable to that of the state-of-the-art methods, which demonstrates the effectiveness of our method.
>
> In the revised manuscript, we have a more intuitive presentation to highlight our contributions. For example, in Figure 1, we added our observations that our model can learn meaningfully samplings of point clouds in different categories, and can detect key regions consistently within a semantic class. In Table 7, we find our multi-scale analysis is consistent with the geometric complexity of 3D shapes.

---

### Official Review · Reviewer_2wxQ · 2021-11-02

**Correctness:** 3
**Technical Novelty And Significance:** 3
**Empirical Novelty And Significance:** 3
**Recommendation:** 6
**Confidence:** 4

**Main Review:**

Strengths:

- The authors propose a novel framework for point cloud processing, which is inspired by the lifting scheme in wavelet transformation. This new approach, equipped with performer and GCN, makes it state-of-art in point cloud classification and segmentation.

- Extensive experiments on the performance of the model, especially w.r.t number of scales, point sparsity and rotations.

- This paper is well written, where readers can easily understand the design of the proposed method, and is clear enough for reproducing the method without any code references.

Weakness:

- Major concern

   1. A lack of consistent motivation of using a lifting scheme.
    Though using the lifting scheme is novel, it is less motivated during the narrative of the paper. The only direct motivation is more or less the Fig.1, where the detail features sort of concentrates on 'junctions'. But it's still hard to interpret, and the number of qualitative results are too limited. I like the lifting scheme story, but I don't think it's being motivated enough through the paper.

    What's the unique edge of using this lifting scheme, apart from the few point gains if you use a voting scheme + performer? Does it learn meaningfully samplings of the original point could? does it highlights consistent keypoints within a semantic class?

   2. Are the comparisons fair?
       I'm not really comfortable comparing the proposed method using the performer. Though it gives a few point in performance, I'm not sure if that's fair for evaluating the lifting scheme against other baselines. This relates closely with concern 1, since I believe the merit of having a lifting scheme should not just be the bold numbers, and it's okay if it's not: it just to be on par with state-of-the-art.

   3. Why more scales hurt?
       I don't really understand why more scale hurts. Why does it introduce `redundant information`? I think it's introducing more and more redundant representation, but not necessarily redundant information. Is it because the more scales you have, the more over-parametrized you are and therefore more prune to overfitting?




- Minor concern

1. Is the name of wavelet transform adequate?
    This may or may not be a huge issue, but it's less intuitive for me to see this as a wavelet transformer, but more or less a lifting transformer. The wavelet in the title sort of suggests there's wavelet filters involved, but it's more like putting neural networks(transformers)  in a lifting scheme.




**Summary Of The Paper:**

This paper proposes a novel 3D point cloud representation learning framework. At the core of this method, is a lifting scheme inspired by wavelet decomposition. The proposed method roughly splits the input data in half at each stage, producing a down-sampled approximation C and detail d. Then C is further processed by the next layer, forming a multiscale pyramid. In summary, the contribution of this paper is:

1. Proposed to use the lifting scheme in point cloud processing, using graph convolution networks and transformers as backbone.

2. Evaluated the method against state-of-the-art baselines and showed that the proposed scheme performs well.


**Summary Of The Review:**

In summary, this paper proposes a method for processing point cloud using a lifting scheme. The proposed method is novel, and with some bells and whistles, it achieves the state-of-art performance on point cloud segmentation and classification. However, there's some major issues blocking me from rating this paper above the acceptance bar. First of all, it is not super clear to the reader why one would like to incorporate the lifting scheme for point cloud processing. I don't think it is just for a few points gain in the performance. Secondly, there's some issues in the evaluation due to the choice of the performer vs plain transformer.

I'm willing to raise my score if the authors could address those concerns.

------------------------

After reading the response from the authors, I would like to raise my score from 5 to 6. The authors have addressed my concerns, and I think this paper meets the bar of ICLR.

---

> ### Author Response · Authors · 2021-11-23
> **Response to Reviewer 2wxQ - Part 1**
>
> **Q1**. A lack of consistent motivation of using a lifting scheme.
>
> **A1**. Thanks for your valuable comments. We apologize that our motivation was not clearly presented in the paper. We will address this critical issue in the revised manuscript. In general, our proposed method is motivated by the following observation. **1.** The geometric components with different frequencies in 3D shapes contain distinct geometric characteristics (e.g., high frequency components correspond to edges / sharp parts, while low frequency components correspond to flat areas / smooth parts) and provide complementary geometric information to the shape representations. To effectively learn 3D shape representations, it is desirable to separate these components and process them differently, rather than feed the whole shapes into a single processor. It, therefore, motivates us to decompose 3D shapes into sub-bands components. **2.** Wavelet analysis in the signal processing field provides a nice framework to decompose signals into sub-bands and thus we design our method based on wavelet analysis. However, the conventional lifting scheme in wavelet analysis adopts hand-crafted linear functions and cannot automatically adapt to data. Therefore, they are only suitable to a limited range of patterns in signals and cannot dynamically adapt to various patterns existing in a collection of point clouds. **3.** These observations motivate us to propose an adaptive lifting scheme, a **data-driven** approach to automatically learn sub-band filter parameters to decompose 3D shapes, which makes our method learnable and be adaptive to different data and tasks. **4.** On the two commonly used benchmarks, the performance of our method is comparable to that of the state-of-the-art methods, which, in our opinion, is able to demonstrate the effectiveness of our newly proposed adaptive wavelet transformer.
>
> In the revised manuscript, we have a more intuitive presentation to highlight our contributions. For example, in Figure 1, we added our observations that our model can learn meaningfully samplings of point clouds in different categories. And can detect key regions consistently within a semantic class. In Table 7, we find our multi-scale analysis is consistent with the geometric complexity of 3D shapes.
>
> **Q2**. The fairness of the comparisons.
>
> **A2**. We agree that the merit of proposing an adaptive lifting scheme is not just to improve the state-of-the-art performance on two almost saturated datasets. The motivation of proposing lifting scheme is to decompose 3D shapes into geometric components with different frequencies for discriminative 3D shape representation learning, as discussed above in **A1**. In our opinion, the comparisons are fair, as the vanilla Transformer (AWT-Net +V (ours)) achieves the performance on par with the state-of-the-art methods, which is able to prove the effectiveness of our method. Thus, we appreciate you could be more tolerant to the marginal performance difference (i.e., 0.3\%). Our intention to use Performer is to speed up the training process, not to boost the performance. Thus we appreciate you could put more weight on our motivation and novel contribution of introducing wavelet transformer that uses sub-bands components for 3D shape representation learning, and be more tolerant to the marginal performance improvement or difference (i.e. 0.1%, 0.3%) on saturated benchmarks (e.g., ModelNet40). If all the reviewers and AC recommend only keeping the vanilla Transofmer and removing Performer in order to make the comparison clearer to highlight the contributions of our wavelet transformer for 3D shape representation learning, we will respect the recommendation to remove Performer to avoid confusion and make it better highlight our contributions.
>
> **Q3**. Multi-scale analysis.
>
> **A3**. We are sorry for missing an explanation for this phenomenon. The Table 7 in the original paper only provides the overall accuracy for shapes from all classes. We have added the per-class accuracy for four object categories in Table 7 in the revised manuscript. We observed that for shapes with simple geometric structures (e.g., radio and wardrobe), more scales have a negative effect on the performance, as more scales make the network prone to over-fitting on these simple shapes. While for shapes with complex structures (e.g., stool and person), more scales are needed to capture both coarse and fine-grained geometric features, which is consistent with our observed results reported in the new Table 7 in the revised manuscript. That is, the accuracy increases with more scales applied to geometrically more complex objects. We find out that the ModelNet40 dataset contains more geometrically simple objects than complex ones, which leads to the overall accuracy drops with more scales used (as we reported previously in Table 7 in our original manuscript).

---

> ### Author Response · Authors · 2021-11-23
> **Response to Reviewer 2wxQ - Part 2**
>
> **Q4**. Minor - Is the name of wavelet transform adequate? This may or may not be a huge issue, but it's less intuitive for me to see this as a wavelet transformer, but more or less a lifting transformer. The wavelet in the title sort of suggests there's wavelet filters involved, but it's more like putting neural networks(transformers) in a lifting scheme.
>
> **A4**. We have clarified the connection between lifting scheme with the conventional wavelet transform in our revised paper as follows: Distinct from hand-crafted wavelet filters which can be expressed in analytic formulas, the lifting scheme proposed in our paper learns wavelets implicitly from data. Once trained, the updater $U(\cdot)$ and predictor $P(\cdot)$ can be regarded as two sub-band basis/filters to subdivide the point cloud into high-frequency and low-frequency components, in analogy to the basis $\psi$ in Eq.(1).
>
> **Q5** .It is not super clear to the reader why one would like to incorporate the lifting scheme for point cloud processing.
>
> Please kindly refer to the response to **A1**.
>
> **Q6**. There's some issues in the evaluation due to the choice of the performer vs plain transformer.
>
> **A6**. Please kindly refer to the response to **A2**.

---

### Official Review · Reviewer_fDxy · 2021-11-02

**Correctness:** 4
**Technical Novelty And Significance:** 4
**Empirical Novelty And Significance:** 4
**Recommendation:** 6
**Confidence:** 4

**Main Review:**

### Strengths:
------------

1. The idea of decomposing 3D shapes into sub-bands components is really nice and to the best of my knowledge such representation has not been explored in prior work.
2. The paper is clear, nicely written and easy to follow.
3. The proposed model outperforms prior work on the object classification task on ModelNet40 dataset and on ScanObjectNN dataset and on the object part segmentation task on the ShapeNet Part dataset.

### Weaknesses:
-------------

1. While I acknowledge that the paper outperforms prior works on both the classification and the object part segmentation task, the proposed model is marginally better than previous methods. For example for the object classification task, the proposed model is only 0.1% better than GDANet on the ModelNet40 dataset, while it performs on par with it on the ScanObjectNN dataset. Therefore, I am wondering how statistically significant are these experiments?
2. GDA-Net by Xu et al. is performing comparably well  with the proposed method. However it is not thoroughly discussed in the Related Work section. Therefore I recommend that the authors describe the differences between their model and GDA-Net in detail so that the reader can easily understand what are the differences between the two models.
3. Various implementation details are missing. For example what are the weights $\lambda_1$ and $\lambda_2$ in Eq (2)? What optimizer did the authors use? For how many epochs was their model trained? What was the learning rate? Unfortunately, I wasn't able to find these details in the main paper. However, the authors should provide them in order for the results of the paper to be reproducible.

### Questions / Detailed Comments:
---------------------------------

1. From Table 1, it seems that the proposed method using the vanilla Transformer performs slightly worse than the variant that uses the Performer. Can the authors provide some intuition for this? I would have expected that using Perform would result in faster inference, however it is not clear to me why it also results in improved performance.
2. What is the number of scales used for the classification accuracy experiment. In Table 1, the proposed model achieves 93.9 accuracy, however from the ablation study in Table 7, the best performing model achieves 93.4 with 2 scales. So how under which configuration was the performance of Table 1 achieved?
3. In prior work, authors typically consider the shape classification task on the S3DIS dataset that contain indoor scenes. I think that the claims of the paper would be even stronger if the authors added an additional evaluation on the more challenging S3DIS dataset.
4. For the Object Segmentation experiment, in Table 3, the authors should also include the results from Point Transformer by Zhao et al. that achieves 83.7 class mean IoU and 86.6 instance mean IoU, as stated in the original paper.
5. For the Even-Odd Split operation, it is not clear to me, why it is necessary to have both a weighted and an unweighted adjacency matrix. Can the authors please comment on this?
6. Some additional references are missing in the Transformers in Vision Section:
- In addition to PCT that applies transformers for 3D pointcloud classification and segmentation, the authors should also mentions the more recent works of:
    * Point Transformer, Zhao et al. ICCV 2021
    * PoinTr: Diverse Point Cloud Completion with Geometry-Aware Transformers, Yu et al. ICCV 2021
- In addition to the mentioned works on efficient transformers, the authors should also mention the work of "Transformers are RNNs: Fast Autoregressive Transformers with Linear Attention" by Katharopoulos et al. ICML 2019, since it was among the first to address the quadratic complexity of transformers

### Minor Comments / Typos:
-------------------------

- In Page 2, in Section Introduction "state-of-the-arts": should be "state-of-the-art"



**Summary Of The Paper:**

This paper proposes a new 3D shape representation learning method using multi-scale wavelet decomposition. In particular, the authors introduce a neural network architecture that decomposed 3D shapes into sub-bands components at multiple scales. In particular, starting from a pointcloud the proposed model learns to decompose it into coarse (high frequency) and detail (low frequency) components using an adaptive lifting scheme, similar to the original lifting scheme introduced for defining second-generation wavelets. Subsequently, two transformer models are used to refine the coarse and approximate geometry of the 3D shape. The proposed model achieves state-of-the-art results on the shape classification task on the ModelNet40 and the ScanObjectNN dataset and on the part segmentation task on the ShapeNet Part dataset. The concept of using such an adaptive lifting scheme seems to facilitate learning and to the best of my knowledge is novel for the case of shape representation learning.

**Summary Of The Review:**

Overall, I really like this paper. It was easy to ready and nicely written. Moreover, the proposed model outperforms previous methods on the scene classification task on multiple datasets and on the part segmentation task. While the proposed model is marginally better than previous models, I really like the concept of utilizing wavelets for 3D shape modelling thus I vote for accepting this paper.

---

> ### Author Response · Authors · 2021-11-23
> **Response to Reviewer fDxy - Part 1**
>
> **Q1**. The proposed model is marginally better than previous methods. For example for the object classification task, the proposed model is only 0.1\% better than GDANet on the ModelNet40 dataset, while it performs on par with it on the ScanObjectNN dataset.
>
> **A1**. Our contribution is to propose a novel framework to learn interpretable representations of 3D shapes, i.e., disentangle 3D shapes into high and low frequency components in a hierarchical manner for multi-resolution analysis. Our framework is inspired by wavelet analysis which has widely been used for signal processing but has not been explored in 3D vision. The performance is on par with the state-of-the-art methods on the two widely used but saturated benchmarks, which, in our opinion, is able to prove the effectiveness of our method. We also notice that the 3D classification task has been almost saturated on these datasets, and thus we sincerely appreciate you can be more tolerant to the marginal performance improvement or difference (i.e. 0.1\%, 0.3\%).
>
> **Q2**. From Table 1, it seems that the proposed method using the vanilla Transformer performs slightly worse than the variant that uses the Performer. Can the authors provide some intuition for this? I would have expected that using Performer would result in faster inference, however, it is not clear to me why it also results in improved performance.
>
> **A2**. Due to the quadratic time complexity of the attention module proposed in the vanilla Transformer, our initial motivation is to adopt Performer to speed up the training process. The results showed that Performer performs slightly better (0.4\%) than the vanilla Transformer. While we do not have theoretical cues to support this observation, one possible explanation is the random sampling process to approximate softmax-kernels for attention in Performer alleviates the over-fitting phenomenon. Importantly, we noticed that with the vanilla transformer, we can also achieve comparable performance with the state-of-the-art methods.
>
> **Q3**. GDA-Net by Xu et al. is performing comparably well with the proposed method. However it is not thoroughly discussed in the Related Work section.
>
> **A3**. Thanks for bringing our attention to this nice paper. We have cited GDA-Net in our original paper and will discuss it thoroughly in the revised version. There are the two main differences between GDA-Net and ours as follows. **1.** The theories based on which we can decompose 3D shapes into sub-bands with different frequencies are different. GDA-Net is built on *graph spectral theory*. GDA-Net uses a 2-order polynomial approximation (Laplacian) as the filter and applies it to a graph to disentangle the graph into high and low frequency components. However, our method is inspired by *wavelet transform theory* which is different from the spectral graph theory and never touches any concept of eigendecomposition. **2.** The decomposition of shapes into high and low frequency components in GDA-Net is *deterministic and static* and independent of tasks. Specifically, given graphs constructed from point clouds, Eq.(5) in GDA-Net separate the high and low frequency components by graph spectral analysis. In contrast, our process of decomposing 3D shapes into sub-bands is adaptive to data and tasks. Specifically, the disentangle of sub-bands components is determined by the task-dependent term (e.g., cross-entropy for classification) in the loss defined in Eq.(10). By varying the network structure of the updater $U(\cdot)$, predictor $P(\cdot)$, and $\lambda_1$, $\lambda_2$ according to different tasks, the decomposition is dynamic and adaptive. Please refer to the related work in the revised manuscript.
>
> **Q4**. What is the number of scales used for the classification accuracy experiment. In Table 1, the proposed model achieves 93.9 accuracy, however from the ablation study in Table 7, the best performing model achieves 93.4 with 2 scales. So how under which configuration was the performance of Table 1 achieved?
>
> **A4**. Sorry for the unclarity and we have made this more clear in our revised paper. The 93.9 accuracy reported in Table 1 is achieved with 2 scales by voting during inference, and such a voting is a commonly used procedure. In the ablation study section, we did not use voting during inference, as stated in the first paragraph in Section 6 in the original paper.

---

> ### Author Response · Authors · 2021-11-23
> **Response to Reviewer fDxy - Part 2**
>
> **Q5**. Various implementation details are missing. For example what are the weights $\lambda_1$ and $\lambda_2$ in Eq (2)? What optimizer did the authors use? For how many epochs was their model trained? What was the learning rate?
>
> **A5**. Thanks for the constructive comments and we have added implementation details in our revised paper.
>
> **Q6**. In prior work, authors typically consider the shape classification task on the S3DIS dataset that contains indoor scenes. I think that the claims of the paper would be even stronger if the authors added an additional evaluation on the more challenging S3DIS dataset.
>
> **A6**.  In prior work, authors often report 3D scene semantic segmentation instead of shape classification on S3DIS dataset. If this is needed, we will report the segmentation results of our model on S3DIS in our final version.
>
> **Q7**. For the Object Segmentation experiment, in Table 3, the authors should also include the results from Point Transformer by Zhao et al. that achieves 83.7 class mean IoU and 86.6 instance mean IoU.
>
> **A7**. We added the shape part segmentation results from Point Transformer in our revised paper.
>
> **Q8**. For the Even-Odd Split operation, it is not clear to me, why it is necessary to have both a weighted and an unweighted adjacency matrix. Can the authors please comment on this?
>
> **A8**. Thanks for pointing out the confusing part and we clarify it as follows: for the even-odd split operation, only the unweighted adjacency matrix is used, as we just need the connectivity between each points in a graph. The weighted adjacent matrix is used in GCN network. However, this is not mandatory, and we can also use the unweighted adjacent matrix for GCN. As GCN is not the focus of our paper, we didn't dive deep into it.
>
> **Q9**. Some additional references are missing in the Transformers in Vision Section.
>
> **A9**. We have added more related papers including the suggested ones in the Transformers in Vision Section in the revised manuscript.

---

### Author Response · Authors · 2021-11-23
**General Responses to All Reviewers**

We sincerely thank all reviewers for the constructive feedback. We appreciate the positive comments on the novelty of our paper, such as "*Decomposing 3D shapes into sub-bands components is really nice and to the best of my knowledge, such representation has not been explored in prior work. While the proposed model is marginally better than previous models, I really like the concept of utilizing wavelets for 3D shape modeling thus I vote for accepting this paper.*'' from Reviewer fDxy. "*The authors propose a novel framework for point cloud processing, which is inspired by the lifting scheme in wavelet transformation.*" from Reviewer 2wxQ. "*Relying on wavelet decomposition for 3D shape learning is theoretically sound and interesting. The experiments are sufficiently demonstrating the effectiveness of the proposed framework design.*'' from Reviewer Dnca. "*Applying wavelet transform for 3D representation learning seems quite new and is not much explored before. For the experiment results, the numbers themselves are stronger.*'' from Reviewer tf1v.

The revised manuscript has addressed the concerns raised by the reviewers, improved the presentation, and added the missing references. Below are the responses to some common concerns.

**Q1**. Motivation and contribution.

**A1**. In general, our proposed method is motivated by the following observation. **1.** The geometric components with different frequencies in 3D shapes contain distinct geometric characteristics (e.g., high frequency components correspond to edges / sharp parts, while low frequency components correspond to flat areas / smooth parts) and provide complementary geometric information to the shape representations. To effectively learn 3D shape representations, it is desirable to separate these components and process them differently, rather than feed the whole shapes into a single processor. It, therefore, motivates us to decompose 3D shapes into sub-bands components. **2.** Wavelet analysis in the signal processing field provides a nice framework to decompose signals into sub-bands and thus we design our method based on wavelet analysis. However, the conventional lifting scheme in wavelet analysis adopts hand-crafted linear functions and cannot automatically adapt to data. Therefore, they are only suitable to a limited range of patterns in signals and cannot dynamically adapt to various patterns existing in a collection of point clouds. **3.** These observations motivate us to propose an adaptive lifting scheme, a **data-driven** approach to automatically learn sub-band filter parameters to decompose 3D shapes, which makes our method learnable and be adaptive to different data and tasks. **4.** On the two commonly used but saturated benchmarks, the performance of our method is comparable to that of the state-of-the-art methods, which demonstrates the effectiveness of our method.

In the revised manuscript, we have a more intuitive presentation to highlight our contributions. For example, in Figure 1, we added our observations that our model can learn meaningfully samplings of point clouds in different categories. And can detect key regions consistently within a semantic class. In Table 7, we find our multi-scale analysis is consistent with the geometric complexity of 3D shapes.

**Q2**. Experiments.

**A2**. We apologize that the design of our experiments that use both vanilla Transformer and Performer somewhat caused the confusion to interpret our true contribution from the experimental comparison, which also leads to the fact that our true contribution of wavelet for 3D shape learning is not fully highlighted well with the relevant results. Using Performer is not a must in our method. It is reported that Performer uses only linear (as opposed to quadratic in the vanilla Transformer) space and time complexity. Our initial motivation is to use Performer to speed up the training process, not intended to improve the performance. As shown in the new Table 11 in Appendix, Performer is faster than the vanilla Transformer.

The performance of our method with vanilla Transformer is on par with the state-of-the-art methods on the two widely used but saturated benchmarks, which, in our opinion, is able to prove the effectiveness of our newly proposed adaptive wavelet transformer. Thus we appreciate the reviewers could put more weight on our motivation and novel contribution of introducing wavelet transformer that uses sub-bands components for 3D shape representation learning, and be more tolerant to the marginal performance improvement or difference (i.e. 0.1\%, 0.3\%) on saturated benchmarks (e.g., ModelNet40). If all the reviewers and AC recommend only keeping the vanilla Transofmer and removing Performer to make the comparison clearer to highlight the contributions of our wavelet transformer for 3D shape representation learning, we will respect the recommendation to remove Performer to avoid confusion and make it better highlight our contributions.

---

### Public Comment · ~Jiaxu_Liu3 · 2023-12-19
**Is Inverse lifting scheme works here?**

Hello,

Thanks for your work.

Can I ask, will you release your code later?

Is it possible to do the inverse transformation of your lifting scheme combined with GCN?  The original lifting scheme seems okay for inverse transformation, but you revise it with optimization with GCN.

---

> ### Public Comment · ~Jiaxu_Liu3 · 2023-12-23
> **Q2**
>
> Is my understanding correct?
>
> when you apply wavelet transformation to a graph, it will generate coarse and detailed coefficients, which correspond to low and frequency points. For example, if we have 100 points in a point cloud, firstly, convert it to a graph, and secondly apply a lifting scheme, then it will generate two groups of nodes which are coarse and detailed nodes, and these nodes sum as 100. This means using a lifting scheme, you separate the nodes into coarse and detailed parts in a regular way.
>
> Even when applying 2-levels lifting scheme, we have d1, c2,d2 (c1), the sum of c2 ,d2 ,d1 nodes is still 100. And c1 has 50 nodes, d1 has 50 nodes, is this correct?

---

### Decision · Program_Chairs · 2022-01-20

**Decision:**

Accept (Poster)

**Comment:**

The submission initially received mixed reviews. The authors presented convincing answers during the author response period, after which all reviewers recommended weak accepts. The AC has carefully read the reviews, responses, and discussions, and agreed with the reviewers' recommendation. Despite the marginal performance gains, the submission has presented a useful and inspiring way of learning shape representations. The AC, therefore, recommends acceptance.

The authors are encouraged to further revise the paper based on the reviews. In addition, the authors should use $\citep$ for all citations that are not used as a pronoun, including all citations in the tables. Please find more information here: https://journals.aas.org/natbib/